



# Life Cycle Assessment of New Jersey Offshore Wind

Meghann Smith[1], Nawal Shoaib[3], Pankaj Lal[1,2]

[1] Department of Earth and Environmental Studies, Montclair State University, 1 Normal Avenue, Montclair, New Jersey 07043, USA

[2] Clean Energy and Sustainability Analytics Center, Montclair State University, 1 Normal Avenue, Montclair, New Jersey 07043, USA

[3] Climate Earth, 137 Park Place, Suite 204, Point Richmond, California, 94801, USA

*Correspondence to*: Meghann Smith (smithmeg@montclair.edu)

**Abstract.** As offshore wind gains momentum within US renewable energy goals, New Jersey's ambitious targets for offshore wind development represent a significant opportunity to reduce emissions and transition towards cleaner energy sources. This study presents a life cycle assessment (LCA) of a planned offshore wind farm off of New Jersey's coast, emphasizing the implications of a domestic supply chain. Key findings suggest that the offshore wind farm is projected to produce 0.013 kg $CO_2$ per kWh of electricity generated, reflecting a 98% decrease in carbon emissions compared to natural gas derived
electricity. Further, when compared to carbon emissions of other renewable energy technologies, offshore wind outperforms both solar and onshore wind by 77% and 39%, respectively. This finding highlights offshore wind's role in greenhouse gas (GHG) emissions reduction through decarbonizing the electricity generation sector. This role is reinforced through the case of a domestic supply chain, a necessary factor in mitigating transportation-related impacts, like fuel combustion, for decreasing emissions. Beyond GHG emissions, results indicate that steel-intensive materials used in turbines and infrastructure contribute
heavily to toxicity-related impacts highlighting a need for seeking alternative, lower-impact materials. This research underscores the potential of offshore wind to reduce greenhouse gas emissions, and offers insight into the environmental dynamics and improved environmental impact based decision making to improve offshore wind deployment in the US.

## 1 Introduction

### 1.1 Offshore wind energy in New Jersey

Globally, offshore wind has been identified as a key player in mitigating the effects of climate change by reducing reliance on fossil fuel-based energy generation. In 2021, the United States (US) federal government announced the ambitious goal of deploying 30 gigawatts (GW) of offshore wind by 2030, and 110 GW by 2050 (The White House, 2021; US Department of Energy, 2022). In line with these national goals, New Jersey set the offshore wind goal of 7.5 GW by 2035, and 11 GW by 2040 (New Jersey Executive Order No. 92, 2019; New Jersey Executive Order No. 307, 2023). New Jersey's current electricity
generation profile does not meet the state's needs; the state produces 64.4 terawatt-hours (TWh) of electric power per year (mostly from natural gas and nuclear sources), but consumes 74 TWh (EIA.gov, 2024). Offshore wind development opens the opportunity to displace the state's reliance on non-renewable sources of electricity generation, improve energy security and independence, and contribute to the national goals by improving the electricity generation mix countrywide.

At the Federal level, Bureau of Ocean Energy Management (BOEM) has facilitated the lease sales of outer continental shelf (OCS) blocks for the development of offshore wind energy farms. Power densities of approximately 3-5 megawatts per square



kilometer (MW/km$^2$) have been estimated for New Jersey's coastline, making this region ideal for offshore wind development (Hernando et al., 2023). There are currently three offshore lease areas with projects in development off the coast of New Jersey (Figure 1). Collectively, these lease areas and their proposed projects will have the capacity to produce 5.2 GW of clean energy.

It is estimated that these projects would cut the state's electricity generation greenhouse gas emissions by 6 million tons annually by displacing fossil fuels (nj.gov, 2024). Supporting these and future offshore wind projects is the New Jersey Wind Port - an offshore wind marshaling port designed to provide open access to the Atlantic Ocean and offshore wind lease areas, and host the technical tradesmen and workforce needed to support the industry.

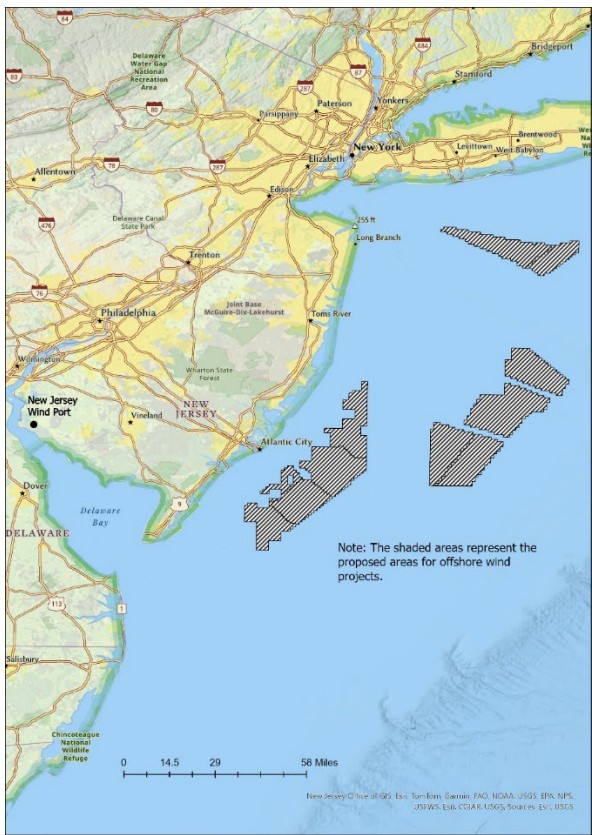

**Figure 1. Map of offshore wind project lease areas and New Jersey Wind Port**

While progress has been made towards reaching these goals, there are several barriers towards the sustainable development of offshore wind energy. One major concern is the limited domestic supply chain needed to support these projects. US Department of Energy (DOE) has established the Near-term Offshore Wind (NOW) initiative which outlines their research and

development (R&D) efforts and plans to address the supply chain issue, among other concerns (US Department of Energy, 2023). Among these R&D efforts include two National Renewable Energy Laboratory (NREL) reports which discuss the demand for and a road map towards a domestic offshore wind supply chain (Shields et al, 2022; Shields, et. al, 2023). In



addition to the New Jersey Wind Port, several manufacturing facilities for critical offshore wind energy components have been announced along the Atlantic coastline which will help to support offshore wind development in the state. In order for US

offshore wind to be successful, and financially viable, this supply chain must exist to meet the demand pipeline and maintain a sustainable industry. Relying on the more established international supply chain has already led to unfortunate circumstances, including delayed timelines which leads to challenges like financial losses, project drop-off, and increased inflation (National Offshore Wind Research and Development Consortium, 2021; Shields et al., 2023).

### 1.2 Scope 1, 2, and 3 emissions

The Greenhouse Gas (GHG) Protocol first defined scope 1, 2, and 3 emissions as a way of categorizing the different kinds of carbon emissions a company creates in its own operations and in its value chain (The Greenhouse Gas Protocol, 2004). Scope 1 emissions are from sources that the organization directly owns or controls. Scope 2 emissions are from the energy purchased in order to support the organizations operations. Scope 1 and 2 emissions are more straightforward to calculate, because the reporting organization has the primary data required in order to calculate the associated emissions. Scope 3 emissions

encompass all other emissions that are not produced by the organization itself, but rather the wider value chain. These scopes are the basis for GHG reporting, and have been widely adopted for mandatory standardization on how organizations measure their emissions (The Greenhouse Gas Protocol, 2015). Because of dynamic supply chains and limited transparency across organizations, the indirect Scope 3 emissions can be incredibly challenging to calculate. It is estimated that over 70% of an organization's emissions could be categorized as Scope 3, making this a crucial component of environmental accounting (The

Greenhouse Gas Protocol, 2011).  By developing an extended environmental impact assessment which includes each scope of emissions, we stand to avoid the concern of "shifting the burden" to another part of the value chain. This circular or systems thinking is critical for accurate emissions accounting as well as sustainable metric design and development.

### 1.3 Life cycle assessment of wind energy

Life cycle assessment (LCA) is a systematic analysis of environmental impact over the course of the entire lifetime of a product,

process, or system. LCA is an advanced modelling technique, compared to traditional input-output models, in that it accounts for the interacting systems in a production value chain. LCA goes beyond traditional $CO_2$ accounting, and expands to include a suite of emissions and aggregates emission data into impact categories, which is useful in providing valuable data for supporting sustainability initiatives. As LCA provides information to evaluate the environmental efficacy of the value chain, it is also useful for hot-spot analysis which can lead to continuous improvement measures in product and process design. LCA

has been used extensively to evaluate renewable energy technology and projects to compare them to traditional fossil fuel-based energy generation, in order to support the justification of transitioning to these often costly and expansive projects.

Recent studies reveal that while offshore wind turbines generally produce significantly lower emissions compared to fossil fuel-based energy sources, their environmental impact can be significant due to the energy-intensive manufacturing of steel





and other materials used in turbine construction. Materials production and transportation stages contribute substantially to the GHG emissions associated with offshore wind, often constituting up to 80% of the life-cycle emissions of a single turbine (Schreiber et al., 2021; Walker et al., 2022; Brussa et al., 2023; Moussavi et al., 2023). Globally, offshore wind life cycle studies have predominantly focused on European markets, where the industry is more established and benefits from the well-developed infrastructure. US studies are however limited; the nuances turbine design, grid integration challenges, limited

supply chain, and specific regulatory requirements are not well understood (Moussavi et al., 2023).

LCA is unique in that geographical context is key for creating a realistic model; while LCAs in different geographies can provide useful insights, they fail to capture the details pertaining to specific geographies and technologies. This research addresses this subject gap by modelling the developing offshore wind industry in New Jersey with an established domestic

supply chain.

## 2 Methods

### 2.1 Goal and scope of the study

This study uses LCA approach which quantifies the potential environmental impacts throughout the life cycle of a system; in this case the system refers to all life cycle stages from raw material extraction to waste management of an offshore wind farm

(ISO 14040:2006, 2018; ISO 14044:2006, 2018). We performed the LCA using SimaPro Version 9.5 software, applying the ReCiPe 2016 method under the hierarchist (H) perspective, which calculates emissions based on global perspective on a 100-year time horizon (Huijbregts et al., 2016). The ReCiPe Midpoint method aggregates complex emission data into 18 cause-impact categories, while the Endpoint reflect damage at 3 areas of protection: human health, ecosystem quality, and resource scarcity (Table S1) (Goedkoop et al., 2009; Huijbregts et al., 2016). These two approaches are complementary in that they

highlight environmental flows with minimal uncertainty, and allows for easier interpretation and relevance of those environmental flows.

This offshore wind farm LCA is meant to act as a baseline model for offshore wind farms in development off the coast of New Jersey, US on the Atlantic continental shelf. The Atlantic Coast has shallower water depths, 60 meters (m) or less, that are

suitable for fixed-bottom offshore wind turbines (US Department of Energy, 2022), as opposed to floating substructures. Atlantic Shores Offshore Wind, LLC was awarded lease area OSC-A-0499 in 2019, and will be launch in two projects (South and North) allowing for a combined 1,510 MW of renewable energy into the state of New Jersey. The Atlantic Shores Offshore Wind South Project is the furthest along in the planning and development phase among all New Jersey offshore wind projects (Atlantic Shores Offshore Wind, Project Information). Designed to operate up to 30 years, the South Project (219.2 km$^2$) will

consist of 105-136 turbines spaced 1.9 km apart connected through inter-array cables, with the most westward point approximately 14 km from the shoreline. There will be two large offshore substations, which connect to the landfall point near





Atlantic City through submarine cables. From the landfall point, the power is transmitted to an onshore substation about 20 km away (Atlantic Shores Offshore Wind, Project Information).

Because construction of this project is not projected to begin until after 2026, this LCA uses the International Energy Agency (IEA) Wind 15-Megawatt Offshore Reference Wind Turbine (IEA Wind 15-MW) with a fixed-bottom monopile support structure to model its offshore wind farm. We assume 105 turbines at a more conservative 25 year lifetime operating at 40% efficiency, with power transmission as specified in the official Atlantic Shores documentation as summarized above. Blades are the turbine component most susceptible to wear over time; however, they are built to last 20-25 years (Lui and Barlow,
2017; Majewski et al., 2022). This model further attempts to construct a domestic supply chain informing the transportation of goods and materials based on locations of announced manufacturing facilities operating on the same regional electricity generation mix (Shields et al., 2023; Smith et al., 2021). The selected functional unit is 1 kilowatt-hour (kWh) of electricity produced by the offshore wind farm to the onshore grid.

**2.2 System boundary**

In the system boundaries of the model, each life cycle stage is included: 1) Materials; 2) Assembly, transportation, and installation; 3) Operation and maintenance; 4) Dismantling, transportation, and end-of-life.

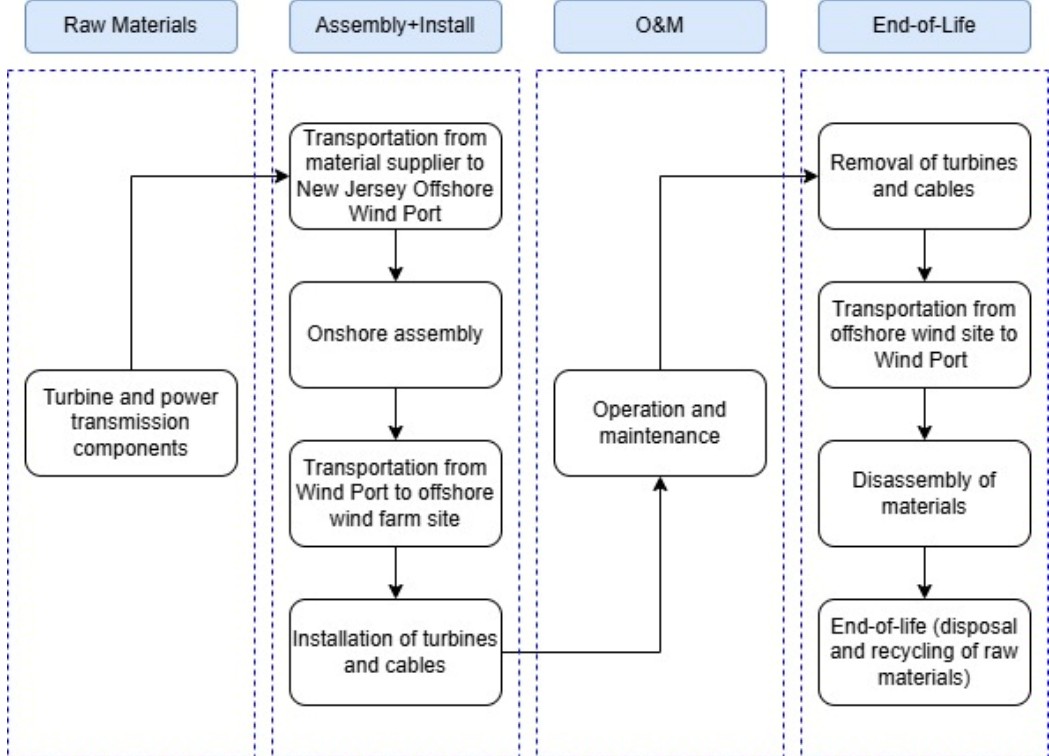

**Figure 2. System Boundary**



Not included in the LCA model is the existing onshore substation, which is used to transform high-voltage electricity so that
it is suitable for local distribution. Local distribution, and end-use of electricity is also outside of the scope of this model as its
considered part of the broader electricity grid infrastructure rather than unique to the offshore wind project itself. While the
offshore substation is included in the model, we limit this to the steel structure and does not include electrical, communications,
or safety equipment in order to focus on components with high material and energy impacts. Assumptions in the transportation
stages includes shortest-distance travelled, both on land and on sea. A more advanced transportation model using weight-
restricted truck routes and sea depth restricted navigation routes can provide additional insights once the exact locations of
these facilities have been announced.

## 2.3 Life cycle inventory (LCI)

The data used in this analysis was collected and adapted from the Ecoinvent version 3.9.1 (allocation, cut-off by classification
– unit) database among other governmental and academic literature sources, where calculations were made such that the model
would be representative of an offshore wind farm as described in the Atlantic Shores Offshore Wind, LLC documentation,
based on the study's functional unit of 1 kWh. The LCI is available in the study's Supplementary Material (Tables S2-S6).

### 2.3.1 Materials

**IEA Wind 15-MW Turbine**
The turbines are modelled off of the IEA Wind 15-MW, which was designed based off the GE Haliade-X 12MW turbine using
a similar drivetrain configuration and specific power, meant to model the potential of offshore wind technology in coming
years (Gaertner, 2020). The IEA Wind 15-MW is an International Electrotechnical Commission (IEC) Class I-B direct-drive
machine, meaning that it is appropriate for high wind speeds (average 10m/second) and low (16%) turbulence, rated for 332
watts per square meter (W/m$^2$).

The tower and monopile are an isotropic steel tube, where the hub height reaches 150 m allowing for 30 m of ground clearance
from the blades, extends 30 m beneath sea level to the mudline. The monopile foundation has a 10 m diameter, and attaches
to the seafloor using an embedded suction pile 45 m below the mudline. While steel makes up the large majority of the mass
(95%), other materials such as those for cables, electronic devices, and lubricant oil are considered in the LCI (Raadal et al.,
2014; Brussa et al, 2023).

The direct-drive nacelle uses a permanent-magnet, synchronous, radial flux outer rotor generator. This design offers several
advantages, including fewer parts, lower complexity, and higher reliability compared to geared drivetrains. The assembly
consists of a hub shaft supporting the turbine, and generator rotors on two bearings which are house on a turret and cantilevered
from the bedplate, and the yaw system connects the bedplate base with the tower top. Based on the details shared in the IEA





report, the nacelle, generator, and rotor are analyzed separately. While these components are primarily made of steel and iron, additional materials are considered including copper, transformers, electronics, and the magnet.

The blade design was based on the rotor diameter of 240 m; 3 117m blades attach to the rotor. The blade design is two main load-carrying carbon fiber spars connecting the root and the tip, with two shear webs that span the vertical length, leading and trailing edge reinforcing glass fiber, and foam filler panels. We assume a 90% of the weight of the blade come from 50/50

carbon fiber reinforced plastic (CFRP) and glass fiber reinforced plastic (GFRP), and 10% polyvinylchloride (PVC) foam.

**Power Transmission**

At its closest point, the South Project wind turbine area is approximately 14 km from the New Jersey shoreline, the turbines will be aligned in a uniform grid, connected through inter-array cables and inter-link cables (totaling about 440 km and 30 km respectively) buried about 2 m beneath the seabed. The inter-array cables are estimated at 35 kg/m, and the inter-link cables

connecting to the substations are estimated at 50 kg/m. The project will require 2 large offshore substations, which will be located approximately 21 km from the landfall point near Atlantic City, estimating four HVDC export cables totaling about 84 km buried about 2 m beneath the seabed (Atlantic Shores Offshore Wind, Appendix I-G), at approximately 60 kg/m. The proposed onshore interconnection cable route (landfall point to existing onshore substation) is estimated at 20 km at approximately 40 kg/m. The material breakdown of power transmission and approximate weight per length of each type was

informed through openAI and previous literature (Brussa et al., 2023; Moussavi et al., 2023; OpenAI, 2023).

**2.3.2 Assembly, transportation, and installation**

Included in the assembly, transportation, and installation phase is the electricity used while operating on the regional grid which is estimated at 50 kilowatt-hours (kWh) per tonne of material (Brussa et al., 2023; Burger and Bauer, 2007), transportation of materials from suppliers to the New Jersey Wind Port (Shields et al., 2023), transportation of materials from

the Wind Port to the wind farm (Atlantic Shores Offshore Wind, LLC, Project Information), seabed transformation (219.2 km$^2$) and occupation (6.58E6 m2y).

**Regional Electricity Generation Mix**

PJM Interconnection is an independent system operator/ regional transmission organization (ISO/RTO) that coordinates the

movement of wholesale electricity in all or parts of 13 states and the District of Columbia. Among these areas covered includes New Jersey, Virginia, and Maryland which are all a part of the modelled regional supply chain described in the following section. We assume the electricity associated with assembly, installation, and disassembly draws upon the PJM Interconnection electricity generation mix, creating a custom process in SimaPro to reflect regionality (Table 1) (PJM.com, 2024).

**Table 1. PJM electricity generation mix**

| Traditional Sources | Percentage | Renewable Sources | Percentage |
| --- | --- | --- | --- |





| Natural gas | 47.02% | Hydro | 1.42% |
|---|---|---|---|
| Nuclear | 29.44% | Wind | 6.33% |
| Coal | 9.64% | Solar | 4.16% |
| Oil | 0.32% | Other renewable | 0.57% |
| Multiple fuel | 1.10% | | |

**Transportation from Supplier to New Jersey Wind Port**

The offshore wind energy industry in the US is gaining momentum, and in order to be viable both in terms of timeline and finances there needs to be a domestic supply chain. While this supply chain is not fully developed, there are several announced

and planned supply chain manufacturing facilities as outlined in the NREL's Supply Chain Road Map (Shields et al., 2023), which is the basis for our supply chain transportation model (Table 2).

**Table 2. Announced domestic supply chain located near New Jersey**

| Location | Approximate distance to New Jersey Wind Port | Supply component |
|---|---|---|
| Portsmouth Marine Terminal, Virginia | 436.13 km | Rotor, Generator, Blades |
| Tradepoint Atlantic and vicinity, Maryland | 149.67 km | Cables |
| Port of Paulsboro, New Jersey | 61.16 km | Tower, Monopile |
| New Jersey Wind Port, New Jersey | 1 km | Nacelle |

The New Jersey Wind Port, is set to be located on the eastern shore of the Delaware River in Lower Alloways Creek, New Jersey. The Wind Port will be the first of its kind in the US, a hub-style marshalling and manufacturing port that will serve wind project in New Jersey and along the US East Coast. The Wind Port will be developed in two phases, 1) a 30-acre marshalling port targeted for completion in 2024; 2) 35 additional acres of marshalling space, enabling two projects to marshal concurrently, and 60-70 acres of space developed for supply chain manufacturers (i.e. Tier 1 components such as nacelles).

Transportation of the parts to the Wind Port uses the largest capacity freight transport lorry available in Ecoinvent, size class >32 metric tons gross vehicle weight which already includes fuel consumption. The equipment that would be needed to load and unload the material is outside of the system boundary.

**Transportation from New Jersey Wind Port to Offshore Wind Farm**

The total distance travelled from Wind Port to the offshore wind site is estimated at 210 km. The Wind Port's location will require ships to travel past the southern tip of New Jersey before entering open ocean towards the offshore wind lease site. For the purpose of this study, we assume this distance to be 140km each way, estimated using GoogleEarth measurement tool



(Google LLC, accessed July 2024). A transport barge is a large flat-bottomed vessel used to transport materials from the port, through inland waterways, and out to the point of installation. To represent this vessel, we chose the freight carrier with a
50,000 ton load capacity. A towing tugboat is needed to tow large vessels safely in and out of inland waterways (Delaware Bay) to/from the New Jersey Wind Port. With no representative vessel available in the Ecoinvent database, we calculated the diesel needed to operate a tugboat estimated at about 150 gallons per hour (WeeksMarine, 2024). Travelling a distance of 70km at approximately 8 knots, the voyage would take approximately 5 hours each direction, consuming 1500 gallons or 5.2 tons of diesel fuel each trip. Due to the load capacity of the transport barge, this would require 8 trips total.

**Installation**

Installation of the offshore wind farm requires several pieces of specialized equipment, including heavy lift vessels, jack-up vessels, dredgers, and service vessels for the crew and operators. While the vessels are not represented in this analysis, fuel consumption of the vessels is included.

The heavy lift vessel is equipped with heavy equipment like specialized cranes needed to move the materials from the barge to the point of installation (consumes 14 tons of diesel fuel per day). The jack-up vessel is a barge with legs that can create a stable platform above the water, used for drilling, dredging, and wind turbine installation (consumes 14 tops/day) (Fred Olsen Windcarrier, 2019). The dredger (18 tons/day) is equipped with excavation tools to remove sediment from the seabed for cable
laying, and the fall pipe vessel (14 tons/day) is used to move cement or rocks to bury the cables after they are installed (De Cuyper et al., 2021; Van Oord 2021). For each of these vessels, a towing tugboat (5.2 tons/day) would be needed to navigate narrower waters (70km) (WeeksMarine, 2024). Service operation vessels (9 tons/day) are used to provide safe extended housing for crew during periods of construction and maintenance (WeeksMarine, 2024). Assuming a 2-year construction period, these vessels would consume about 55,000 tons of diesel fuel combined. Not included, is the use of helicopters during
installation, maintenance, and disassembly, which in some cases are necessary to support transporting and hoisting personnel and equipment to save time especially during challenging weather conditions.

### 2.3.3 Operation and maintenance

Offshore wind farms require corrective, preventative, and predictive maintenance in order to efficiently operate and maintain the machines and their wearable parts. Both South and North parts of the Atlantic Shore Project will be designed to operate
autonomously without on-site technicians, and equipped with supervisory control systems and monitoring sensors to interface between the various components of the site. These remote-control systems are outside of the scope of this work. This LCA assumes regularly scheduled annual maintenance, and no unscheduled maintenance, where crew will inspect, test, replace consumable materials, and complete any preventative maintenance needed. This will include the use of a service operation vessel, and the replacement of lubricating oil for each turbine. We assume that all wearable parts require no material
replacement (e.g. blade removal and replacement) as they are built to last the full 25 year lifetime.





### 2.3.4 Dismantling, transportation, and end-of-life

This process represents the removal of all property and restoration of the leased area on termination of the lease as per the lease agreement. Dismantling the offshore wind turbines is modelled using the same inputs as the "Assembly, transportation, and installation" with a few exceptions. Transportation from New Jersey Wind Port to supplier is not included, land
transformation is from seabed infrastructure to unspecified, and land occupation is to sea and ocean.

This process also includes material end-of-life after it has been disassembled, following the method of Brussa, etal 2023. Metals (steel, aluminium, copper, and iron) are 90% recycled and 10% landfilled. High recycling rates in industry standards are supported by established recycling infrastructures with high recovery rates (IEA.org, 2021). The recycled the process used
is an empty process with no costs or avoided costs to avoid double counting; however, recycling tends to be an energy-intensive process and should be further explored in future research. Glass from the turbine blades and plastics from several components are 100% incinerated. Electronics and cables are 100% treated and disposed, and lubricating oil is 100% treated as hazardous waste and incinerated.

### 2.4 Comparison analysis

To further assess the performance of the offshore wind farm, we compare the model to conventional and renewable electricity generation sources currently used by PJM Interconnection (Table 1). The selected Ecoinvent processes are listed in the Supplementary Materials (Table S7).

## 3 Results and Discussion

### 3.1 Life cycle impact assessment

To calculate the environmental impact of Atlantic Shores South, we included the turbine and power transmission materials, transportation from domestic suppliers to the New Jersey Wind Port, transportation to the offshore wind farm area, assembly and installation of the turbines and cables, annual operation and maintenance, disassembly, and end-of-life waste treatment. The model assumes having 105 fixed monopile turbines (15 MW) connected through inter-array cables and linked to two offshore substations, where export cables run to the shoreline where they connect to the onshore substation, with a 25 year
operational lifetime. The impacts across each midpoint and endpoint category per the functional unit of 1 kWh of wind energy power produced are shared in the Supplementary Material (Table S8).

The results of the impact assessment display the emissions produced by the offshore wind farm per 1 kWh of wind energy power produced. Across several midpoint impact categories, the tower and monopile account for the largest percentage of
impact, primarily due to the large amount of steel (Figure 3). The overall global warming impact of 1.35E-02 kilograms carbon





dioxide equivalent (kg CO$_2$ eq) is substantial, with 34.7% coming from the tower and monopile. Steel production is known to have a significant environmental impact, accounting for approximately 9% of global greenhouse gas emissions. The coke ovens used for steel production are also associated with high air pollution, releasing emissions such as naphthalene and sulfur during the cooking process. Steel production also produces substantial wastewater; the overall water consumption is estimated at 1.13E-04 cubic meters (m3), with 27% coming from the tower and monopile process.

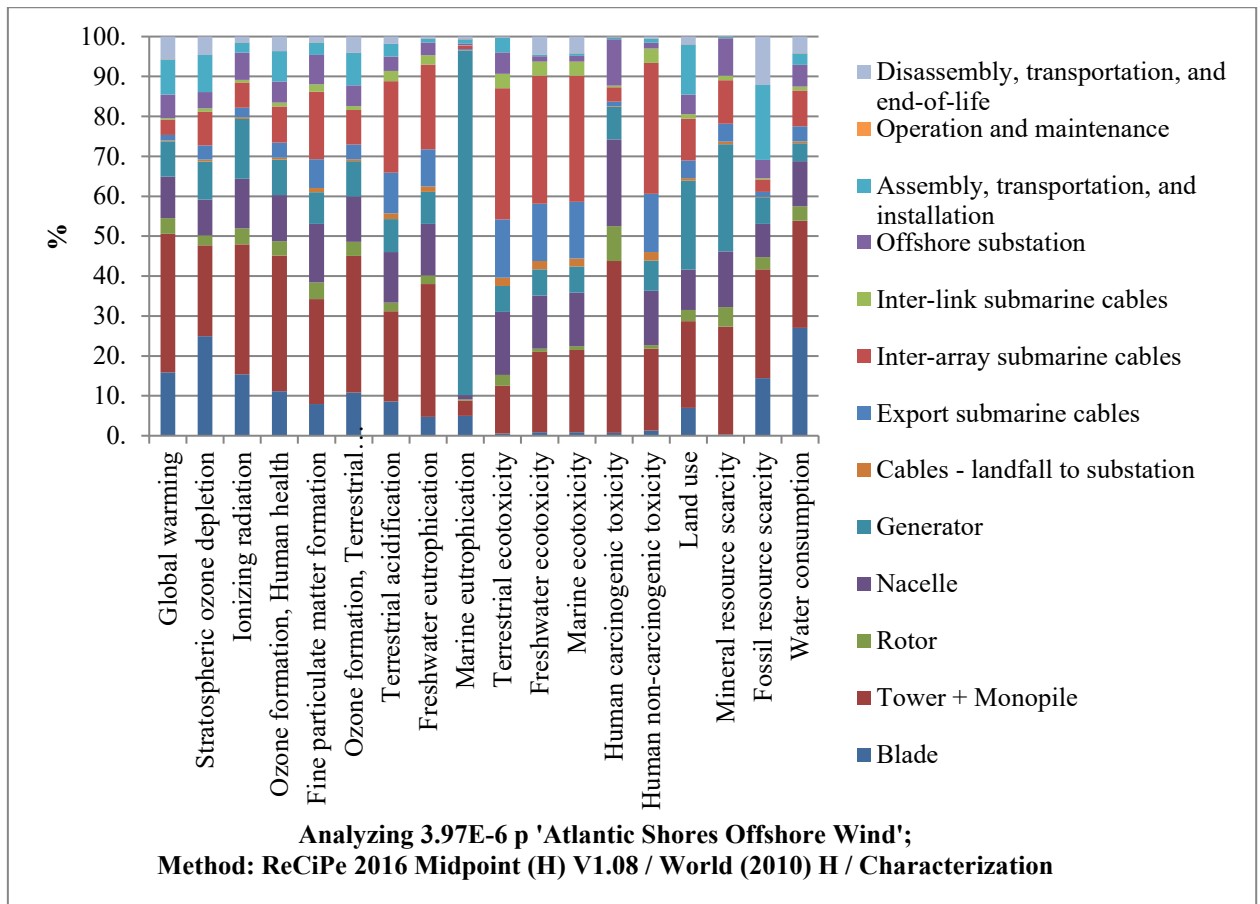

Analyzing 3.97E-6 p 'Atlantic Shores Offshore Wind';
Method: ReCiPe 2016 Midpoint (H) V1.08 / World (2010) H / Characterization

**Figure 3. Life Cycle Impact Assessment – Midpoint Results**

While CO$_2$ is the most commonly reported emissions, it is often not the most important in terms of environmental and human health impact. Normalization of the data allows the impact category indicator results to be compared to a reference (normal) value (Normalization scores ReCiPe 2016, 2020). This allows for perspective of priority areas across the several impact categories. With normalization, we see that human carcinogenic toxicity is by far the highest concern, with an overall impact of 1.15E-02 kg 1,4-dichlorobenzene (1,4-DCB). Again, we see that the tower and monopile process are the largest contributors at 43.1%. Other notable contributors include the nacelle (21.7%) and offshore substation (11.7%) due to their large amount of steel. Freshwater (5.11E-03 kg 1,4-DCB), marine (6.64E-03 kg 1,4-DCB), and terrestrial (4.63E-03 kg 1,4-DCB) ecotoxicity





are also shown to have substantial impact compared to the other categories. While the tower and monopile process is a major contributor, the inter-array submarine cables show the highest impact across each of these categories, at 32.1%, 31.6% and 32.9% respectively.

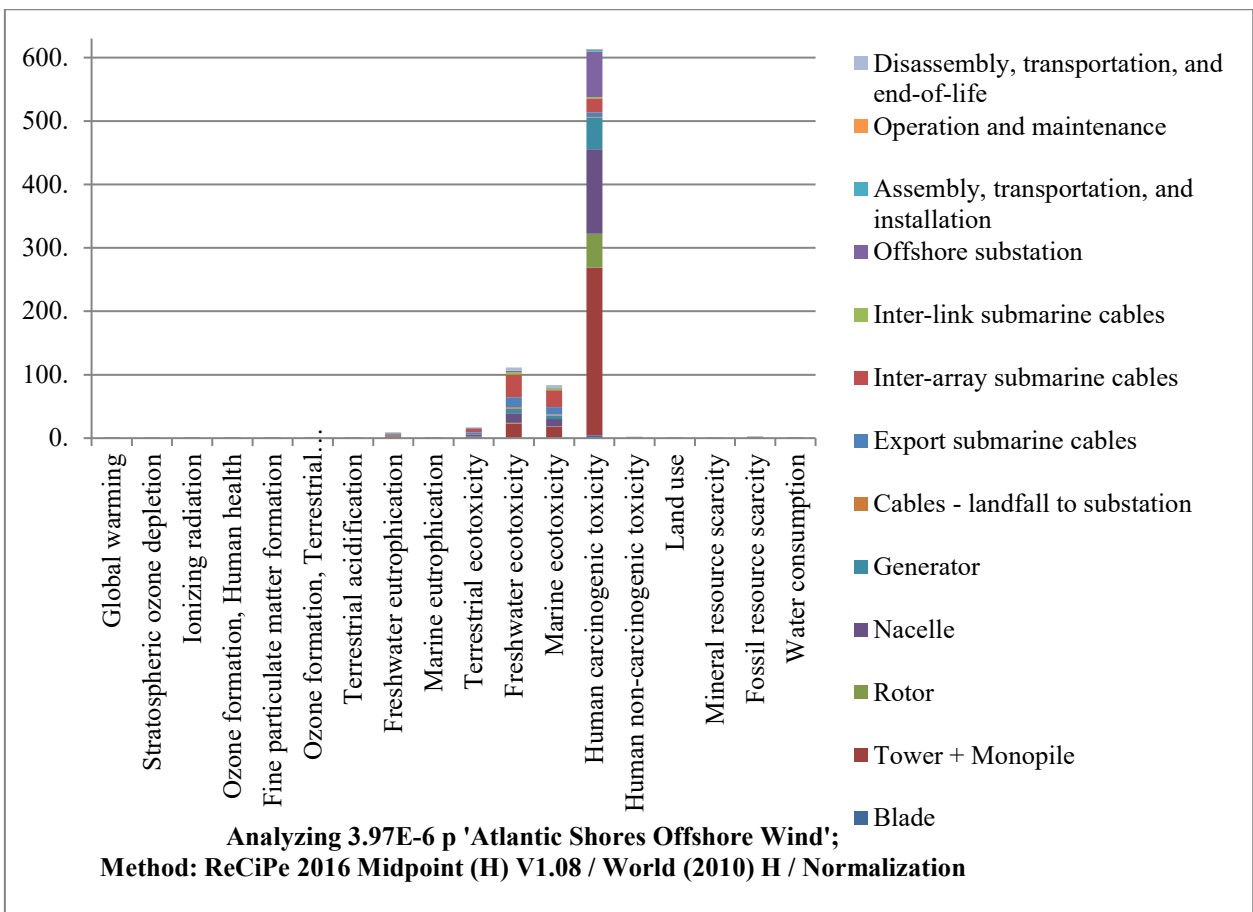

**Figure 4. Life Cycle Impact Assessment – Midpoint Results with Normalization**

The endpoint approach further aggregates the emissions data into three impact categories: human health, ecosystems, and resources. The overall human health impact is 9.19E-08 disability adjusted life-years (DALY), with the tower and monopile having the largest contribution at 33.3%. The overall ecosystem impact is 8.22E-11 species per year (species.yr), with the tower and monopile having the largest contribution at 28.7%. The overall resource impact is $0.001 (USD-2013), with the assembly, transportation, and installation phase having the largest contribution at 24.2%. On further investigation, we see that this is primarily due to the diesel fuel needed to transport materials from the New Jersey Wind Port to the offshore wind farm (64.2%), and the transportation of materials from the suppliers to the Port (35.5%). This finding further highlights the necessity of a domestic supply chain in terms of human and ecosystem health, as well as the impact on the economy.





Figure 5. Life Cycle Impact Assessment – Endpoint Results

The normalization factors used for ReCiPe 2016 endpoint method will always prioritize human health, as seen in the normalization of the data (Figure 6). While fuel consumption and the associated emissions is a key area of concern in regards to sustainability metrics, we see that steel must be prioritized in light of a growing domestic supply chain.



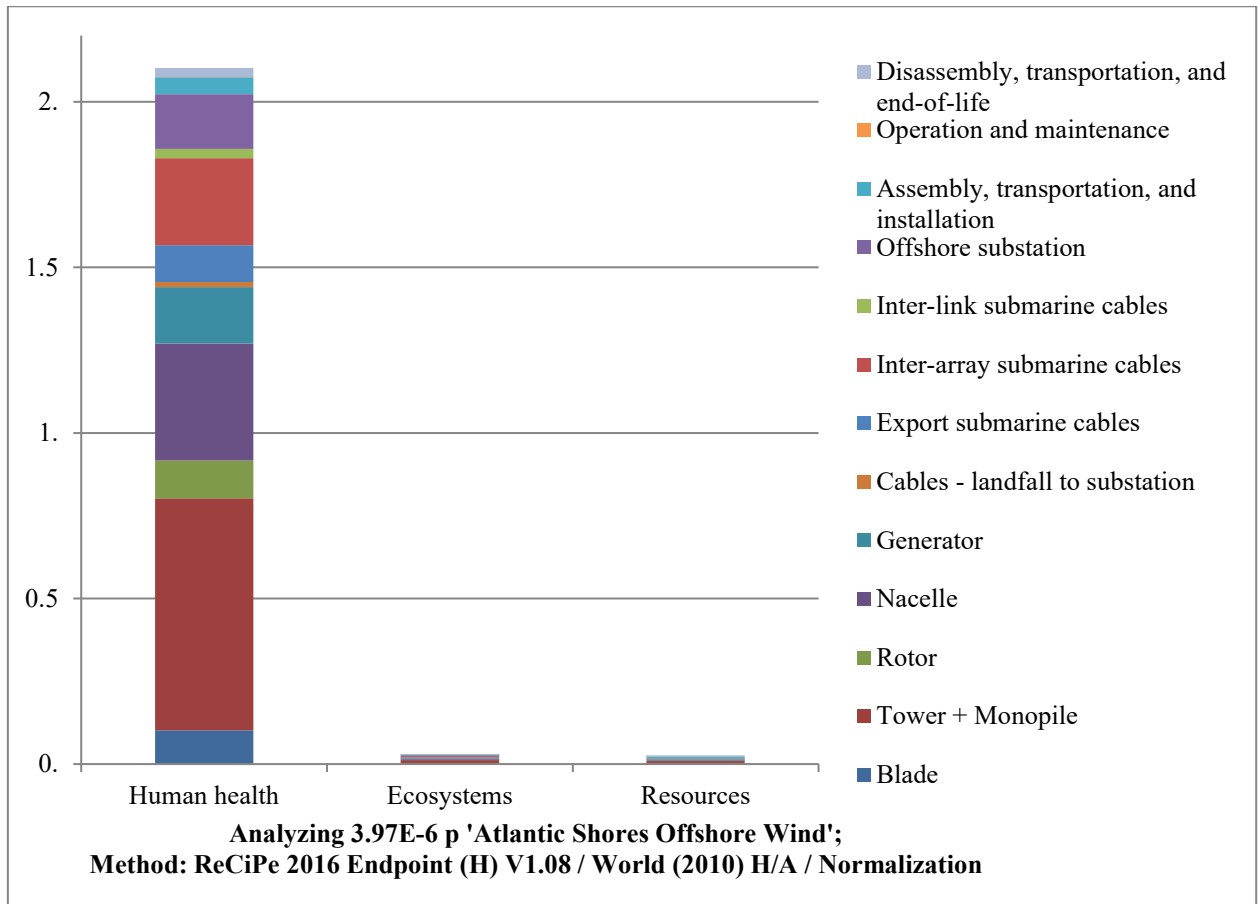

Analyzing 3.97E-6 p 'Atlantic Shores Offshore Wind';
Method: ReCiPe 2016 Endpoint (H) V1.08 / World (2010) H/A / Normalization

**Figure 6. Life Cycle Impact Assessment – Endpoint Results with Normalization**

The results from this study shows similar distribution trends compared to prior offshore wind based LCA studies. Two studies have used the IEA reference turbine as the basis for their LCA model; however, the system boundaries are not directly comparable. Brussa et al. (2023) estimated 0.31 kg $CO_2$ eq per kWh for a 190 turbine farm off Italy's coast; while this is substantially higher than our model's estimate, this study used floating foundations which require even more steel for the floating substructure and mooring system. Moussavi et al. (2023) reports that 62% of $CO_2$ emissions are associated with the tower, offshore substation and cables, the most steel intensive categories which mirrors our findings of about 70% when including the nacelle, generator, and rotor which were excluded from their report. Because this study reported their normalized values, we are unable to directly compare results. While several other studies have evaluated offshore wind, the turbine power rating is significantly lower such that results cannot be directly compared.





## 3.2 Comparison analysis

The US EIA estimated the resulting $CO_2$ emissions by fuel source, including petroleum (1.08 kg $CO_2$ per kWh), coal (1.04 kg
$CO_2$ per kWh), and natural gas (0.44 kg $CO_2$ per kWh). Compared to our finding, it is clear that offshore wind technology for electricity generation is the clear path for decarbonization (EIA.gov, 2023). Further, because the US operates on a diverse electricity generation mix, including 21% of electricity produced from renewable sources, the same report estimates that the nation's grid produced 0.39 kg $CO_2$ eq per kWh of electricity produced in 2022 (EIA.gov, 2022). Continuing the push for electricity generation from renewable sources is key to reducing this number as a nation, and offshore wind allows New Jersey,
a small state with high energy needs, to participate in improving national carbon emissions.

Using Ecoinvent process data, we evaluated the performance of other electricity generation sources used by the PJM Interconnection electricity generation mix across impact categories beyond global warming potential, separated for ease of visualization by conventional (Figure 7) and renewable (Figure 8) sources of electricity generation. The full results of the
analysis are shared in the Supplementary Material (Table S9).

The $CO_2$ emissions from the three above mentioned sources were somewhat different at 2.05 kg $CO_2$ eq (oil), 1.02 kg $CO_2$ eq (coal), and 0.62 kg $CO_2$ eq (natural gas); without full details of the EIA estimation process it is difficult to know why the estimations for oil/petroleum are so different. We expand our analysis to include nuclear (pressurized water reactor [PWR]
and boiling water reactor [BWR], and we see that these two approaches tend to outperform offshore wind in terms of $CO_2$ emissions. While these are clean electricity technologies, there other impacts such as ionizing radiation highlight the risk of relying on nuclear for a clean energy transition. Compared to conventional sources, we see that offshore wind performs very well across most environmental impact categories. Results confirm, however, that the impact off offshore wind to freshwater, marine, and human carcinogenic toxicity categories remains a concern compared to several conventional methods of electricity
generation with the exception of coal and oil. We also see the impact on mineral resource scarcity remains relatively high compared to coal and natural gas. Each of these is related to the large amount of steel needed for the turbines, highlighting the importance of identifying alternative materials for these massive structures.

When analysing the renewable electricity generation sources, we see that in terms of $CO_2$ eq, offshore wind outperforms solar
and onshore wind by 77% and 39%, respectively. Hydroelectricity outperforms offshore wind by 66% (run of river) and 48% (pumped reservoir) depending on the method. The only impact category where offshore wind performs the worst is marine eutrophication, which makes sense based on location of the wind farm. Other notable impact categories where offshore wind compares poorly to several other renewable technologies include human carcinogenic toxicity and mineral resource scarcity, which is again linked to the impact of steel. Overall, in this analysis it appears that hydroelectricity has the lowest impact
across most impact categories. However, the effect of hydroelectric dams on water quality and habitat destruction should not



be considered lightly. Further, hydroelectric dams can only be utilized in certain geographies; while New Jersey has several hydroelectric dams in operation possibilities of expansion in this sector are minimal given land limitations.

**Figure 7. Comparison of offshore wind to conventional sources of electricity generation**


**Figure 8. Comparison of offshore wind to renewable sources of electricity generation**

## 4 Conclusions

This LCA of Atlantic Shores South offshore wind project off New Jersey's coastline provides a detailed analysis of the environmental impacts associated with offshore wind energy production in the context of a developing US supply chain. Further, we compare these results to other electricity generation sources including traditional (oil, coal, natural gas, nuclear) and renewable (solar, hydro, onshore wind) technologies. Findings suggest that while offshore wind offers a substantial reduction in greenhouse gas emissions relative to fossil fuels, there are considerable environmental impacts associated with the steel-intensive construction materials and transportation of those materials, particularly within the Scope 3 emissions profile. Scope 3 emissions, which include indirect emissions from upstream activities such as material production and transport, are found to be a major contributor to the projects total environmental impact. Steel manufacturing, in particular, poses substantial challenges for sustainable offshore wind development due to its high emissions intensity and toxicity impacts





on human health and ecosystems. When we consider performance of offshore wind to other renewable energy technologies, we observe that in general offshore wind is a good option, particularly in the context of New Jersey which has ample coastline

access and limited available land space for expansion of other electricity generation options. This research provides quantitative data that can guide regulatory frameworks, incentive structures, and investment decisions that prioritize both emission reductions and sustainable industry development. By aligning with other renewable technologies' performance, this research also contributes to broader energy transition planning, supporting evidence-based policies that promote the most environmentally and economically viable solutions for the US energy grid.


A lack of a domestic supply chain is a large, well-known barrier to US offshore wind development. The emissions associated with transportation of turbine components overseas highlight the ineffectiveness of utilizing offshore wind as a means to decarbonize the electricity generation sector. Implementing a more localized supply chain would not only reduce transportation-related emissions but also foster greater energy security and economic benefits within the region. This study

contributes a crucial baseline for future US east coast offshore wind LCA models, and emphasizes the importance of regional context in sustainability metrics. New Jersey's offshore wind developments offer a promising step toward renewable energy transition, provided that the challenges of supply chain sustainability and resource management are effectively addressed.

Future research can explore the potential impacts of a decarbonized steel sector, assessing how a transition to low-carbon steel

production would alter the environmental footprint of offshore wind projects. Further modelling efforts could also assess the use of innovative materials, such as composite or recycled materials, and their feasibility within offshore wind applications. By investigating alternative materials, future work could identify best management practices within the offshore wind sector as it continues to develop its domestic supply chain, ensuring that as a nation we are providing a strong foundation for a sustainable offshore wind sector.

**Data availability**

Data for this study is available in the Supplementary Material.

**Author contribution**

MS contributed to the conceptualization, methodology, project administration, formal analysis, validation, and writing – original draft preparation. NS contributed to the conceptualization, methodology, formal analysis, and writing – review &

editing. PL contributed to the conceptualization, funding acquisition, supervision, and writing – review & editing.

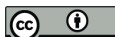



**Competing Interests**

The authors declare that they have no conflict of interest.

**Acknowledgement**

The authors are grateful for funding from the Clean Energy and Sustainability Analytics Center at Montclair State University.
Any opinion, findings, conclusions, or recommendations expressed in this material are those of the authors and do not necessarily reflect the views of the Clean Energy and Sustainability Analytics Center at Montclair State University.

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
