# Peer review of "Life Cycle Assessment of New Jersey Offshore Wind"

_Wind Energy Science, 2024_

## Author Response (AR1)

The authors thank the editor and reviewers for their time and thoughtful input. The responses to the reviewers' comments are written as bullet points. The authors would like to note a change in the results based on a comment from Reviewer #2 which has led to an improved, more accurate representation of transportation of turbine components and cables from the domestic supply chain to the New Jersey Wind Port. All figures and values have been updated to reflect the new results in both the manuscript and supplementary material file.

**Reviewer #1**

**General Comments/Overall Quality**

This paper provides a current perspective on potential emissions reductions of the prospective New Jersey offshore wind leases. Using a developed infrastructure modeling scenario with regional supply chains, energy grids, manufacturing ports, and turbine technical details that have been deemed essential by government agencies and peer-reviewed literature that differentiates this study from other analysis. It is a rational starting point to reference as a framework for future iterative research. However, an important remark should be made pertaining to the political, economic, and temporal constraints that may occur over the next decade or more. The impactful and relevant data are concise and consistent in styling with other peer-reviewed publications. The life cycle assessment methodology is straightforward and conventional without novel allocation or system expansion. The results highlight necessary improvements in production and manufacturing upstream that should be targeted to measurably improve emissions from offshore wind now and in the future.

**Individual Scientific Questions or Issues**

Line 105: It is unclear if the reference to 'minimal uncertainty' is based on the use of ReCiPe itself or translation between midpoint and endpoint. For managing uncertainty, ReCiPe is well regarded for characterization and pollutant transport. Midpoint has lower uncertainty whereas the environmental relevance of endpoint may be of more interest. However, uncertainty may also be influenced by the goal & scope and life cycle inventory, both of which may be discussed briefly also.

- This sentence has been revised to "These two methods.." to improve clarity. Further, I added final sentence to paragraph 2 in section 2.1 to address how goal and scope contribute to less uncertainty, "Relying on the published details of the planned South Project to create the framework for this study makes for a narrow goal with a well-defined scope to reduce uncertainty in the model in a developing sector."

There should be further clarification how OpenAI referenced in Line 179,180 was used, how the data was validated, or deemed of sufficient quality. Use of AI, machine learning, extrapolation, and approximate corollaries in life cycle inventories is often necessary but AI resources may not produce reliable data.

- This statement has been revised to say "…through openAI and validated referencing previous literature" to improve clarity. The previous literature referenced also includes the power transmission as a portion of their system boundary; however, we use copper as the conductive material due to its greater reliability and remove cast iron as the South Project build plan states the inter-array and inter-link cables will be buried 2m subsea so they should not need ballast.

Further explanation or refining Line 259-261 may be warranted on the end-of-life, specifically with respect to concerns about double counting and the energy intensity of recycling. If possible, perhaps cite other literature that has taken the same approach. Designing the process in this manner will not generate

emissions credits via system expansion but may be dismissive to trade-offs. Perhaps energy and labor intensity would be appropriately cited as research elsewhere that focuses on end-of-life.

- The statement has been revised to say "..future research to better assess environmental trade-offs" to make this limitation of the study's method more explicit. The method has been validated in previous work, and references have been added:
- Brussa, G., Grosso, M., & Rigamonti, L. Life cycle assessment of a floating offshore wind farm in Italy. Sustainable Production and Consumption, 134, 134-144. https://doi.org/10.1016/j.spc.2023.05.006, 2023.
- International EPD ® System. General Programme Instructions for the International EPD ® System Version 4.0, 2021.

Section 3.2, Lines 329-330 compares the carbon dioxide per kg emissions results produced from this study with those from the United States Energy Information Administration (USEIA). The fact that the USEIA emissions calculation method is unknown (Lines 341-344) should be stated following this initial comparison. The USEIA estimates have value but those calculated within the PJM network, using ReCiPe, may be more appropriate in the scope of this paper.

- The section details have been reorganized to improve clarity. The details on the US EIA results have been moved to the methods to provide support for completing the comparison analysis, and the results have highlighted the differences between the US grid and PJM electricity generation mixes which we believe is a more appropriate measure to compare our offshore wind model results to.

Section 3.2, Lines 360- 362 it is implied that the impacts to water quality and habitat destruction are externalities unrelated to elemental (chemical) emissions not included in this analysis. This should be made clearer. Analysis of these issues may be better calculated using an environmental impact statement. Similar acknowledgments could be made about implications and risks of nuclear energy including long-term storage/land use loss in Lines 344-347.

- Thank you for the suggestion, these areas have been revised so they read clearer that the model estimates emissions and is not necessarily indicative of the environmental implications of those emissions.

**Technical Corrections**

Line 78: The data or results produced from a life cycle assessment should *inform* sustainability initiatives. It may not unilaterally support them as stated.

- Thank you, the change has been made.

Line 78-79: This may be broken into two sentences.

- Thank you, we kept this as one sentence but revised to explain how LCA is useful for downstream and upstream hotspot analysis since we are referring to the value chain.

Line 79-81: Rephrase or break into two sentences.

- Thank you, we have revised this to say "LCA has been used extensively to evaluate the environmental impact of renewable energy technology and projects. By comparing these newer technologies to traditional fossil fuels, LCA has proven a useful tool for justifying the transition to these often costly and expansive projects.

Line 175: Revise HVDC to High Voltage Direct Current unless it is referenced earlier.

- The revision has been made.

Line 259: There is a grammatical error 'The recycled the process…'

- Thank you, the revision has been made to "The recycling process..".

Line 260: '…no costs or avoided costs' is better referenced as emissions or impact unless there is a further economic implication being discussed in this paper or section.

- Thank you, we revised to "…no emissions or waste…".

Line 307: State the impact category that is influenced by diesel fuel needed. It is implied that it is natural resource scarcity derived from midpoints mineral resource scarcity and fossil resource scarcity.

- I have added the reference to the first sentence of that paragraph Huijbregts et al, 2017 which describes how the midpoint impact categories are aggregated for the endpoint impact categories which will best explain how each human health, ecosystems, and resources are calculated.
- Huijbregts, M.A.J., Steinmann, Z.J.N., Elshout, P.M.F., Stam, G., Verones, F., Vieira, M., Zijp, M., Hollander, A., & van Zelm, R. ReCiPe2016: A harmonized life cycle impact assessment method at midpoint and endpoint level. The International Journal of Life Cycle Assessment, 22(1), 138-147. https://link.springer.com/article/10.1007/s11367-016-1246-y, 2016.

Line 357: '… which makes sense' expects the reader to draw conclusions and is informal. A finite statement can be made that acknowledges the impacts reflect the relative location of these compared technologies.

- Thank you, the sentence has been revised based on your recommendation.

Figure 7, Figure 8: These figures are not a comparison of product stages as stated in the figure. It is a comparison of technologies/scenarios/energy sources.

- The figures have been renamed to state "…to [conventional/renewable] electricity generation technologies"

**Reviewer #2**

General comments

This paper presents a life cycle assessment of the Atlantic Shores offshore wind project in New Jersey. The authors provide clear explanations of their methods, assumptions, and conclusions. The analysis contributes to scientific understanding of offshore wind impacts by considering a new geographic region and adding to the relatively small body of studies of the technology at this scale (15 MW). The authors use well-established methods and provide documentation of their inputs and assumptions. The findings are presented clearly with informative figures and explanation.

Specific comments

Lines 35-36 "Power densities…" The range of 3-5 MW/km$^2$ describes the expected wind turbine spacing and is not indicative of the quality of the wind resource. Wind speed is more typically cited in this context.

- We have revised this line to read, "Average wind speeds approximately 8-9 meters per second at 90 meters high have been measured for New Jersey's coastline, making this region ideal for offshore wind development." and have cited a new reference below.
- U.S. Energy Information Administration. Off Shore Wind Speed 90m (NREL), https://atlas.eia.gov/datasets/ddc4fbdf89b54d63b24ad2f841bf1f86_0/about, last access: 25 March 2025, 2021.

Line 57 "increased inflation" While inflation has been a challenge for offshore wind projects on the East Coast (and elsewhere), it would be more accurate to say that inflation has increased project costs rather than that international offshore wind supply chains have caused inflation.

- We agree, and have revised the sentence to highlight that the issue is more circular, "Dependence on established international supply chains has resulted in a cyclical pattern, where delays in project timelines contribute to financial losses, project cancellations, and escalating inflation, thereby exacerbating both costs and delays."

Lines 122-123 "40% efficiency" Do you mean capacity factor?

- Yes, we have added "(capacity factor)" to this line as we have seen this operational detail described in both ways in previous literature.

Lines 136-138 "While the offshore substation is included…" I understand you're limiting the scope, but some of the electrical equipment in the offshore substation is quite massive and may contain substances with high environmental impact potential, for example, SF$_6$ in some gas-insulated switchgear. Have you done any analysis of how including substation equipment could affect your results?

- Thank you for the comment, the authors agree that this is a needed area for exploration; however, given the scope of the research and data availability we were unable to model the substations. We have added this limitation to the Conclusions section highlighting this area of interest in future research pursuits. "Future research should explore the aspects of offshore wind farms that are outside of the scope of this study. For example, the offshore substation contains substantial materials beyond steel, such as transformers, high voltage equipment, power control systems, and

communication equipment, which could have a significant environmental impact relative to the total offshore wind farm."

Lines 210-212 "Transportation" Why was land transportation assumed between the manufacturers and the NJ Wind Port? Major offshore wind components are too large for conventional land transport, which is why all the selected manufacturing sites are located at ports. Ship transport would be a more appropriate assumption here.

- We thank the reviewer for this note, it was an oversight in our model. We have taken new measurements using the same method (Google Earth measurement tool) to estimate the transportation distance via water from each of the supply chain ports and used the sea freight carrier for shipment + tugboat (fuel consumption only) for inland waterway navigation to the NJ Wind Port. The details of these calculations and values are shared in the methods section of the report. The figures and numbers in the manuscript text have been updated to reflect the improved model, as has the supplementary material file.

Lines 246-250 "This LCA assumes regularly scheduled annual maintenance…" Eliminating all unscheduled maintenance and major replacements is not very realistic. It would be helpful to provide readers with an idea of how much this could affect results, for example Allekotte and Garrett (2024) "Life Cycle Assessment of electricity production from an offshore V236-15 MW wind plant" finds that doubling or halving replacement parts changes impact indicators by <2%.

- We agree, the operation and maintenance process is not realistic; however, we decided to not include emergency maintenance and parts replacement because the calculations in previous literature are inconsistent (from 5%-33% of the full turbine over its lifetime was noted while designing the study). We have acknowledged this limitation in the results section, and have cited the reference you suggested. "Notably, across several impact categories "Operation and maintenance" has a relatively low impact. This model is limited to annual transport of lubricating oil during scheduled maintenance. Unscheduled and emergency maintenance, and replacement of turbine components could greatly change the impact of this category. Allekotte and Garrett (2024) found that when doubling replacement parts, the impact on all categories increased in the range of 0.2-2.1%.
- Allekotte, L, Garrett, P. Life Cycle Assessment of electricity production from an offshore V236-15 MW™ wind plant. Vestas, version 1.1, 2024.

Lines 308-309 "This finding further highlights…" This conclusion reaches beyond the evidence presented here. Without comparative data for domestic and international supply chains, there is no support for the statement that one has lower impacts.

- We have deleted this line, and instead made the statements, "Result suggest that even with a domestic supply chain, the impact of transporting materials is significant, especially in the form of fossil resource consumption. Greater distances travelled from an international supply chain would yield even higher impacts, making a case for a domestic supply chain as well as exploration into alternative fuel use."

Lines 376-378 "Steel manufacturing…" Given the importance of steel manufacturing in your results, it would be informative to discuss the relative impacts of steel produced by electric arc furnace (predominant in U.S. steelmaking) versus basic oxygen furnace (represented in the Ecoinvent Steel, low-alloyed, hot rolled {GLO} dataset).

- We agree, and have added the following to the Results and Discussion section of the paper, "The steel process used in this study is produced by basic oxygen furnace, which is highly carbon intensive and relies on substantial amounts of water for cooling and emissions control. However, the more common method of steel production in the US is by electric arc furnace (EAF), which has a lower environmental impact due to its greater use of steel scrap and less water consumption (American Iron and Steel Institute, 2023). Future work modelling the EAF steel production pathway is likely to present lower emissions and a more accurate outlook on a fully domestic supply chain."
- American Iron and Steel Institute. Steel Profile. https://www.steel.org/wp-content/uploads/2024/01/AISI-Profile-Book_updated-3.2023.pdf, last access 35 March 2025, 2023.

Technical corrections

Line 233 "consumes 14 tops/day" should be tons/da

The correction has been made.

---

## Referee Report (RR1)

The authors' revisions to this manuscript have addressed the issues identified in the initial review.